# Antibiofilm Effect of Cinnamaldehyde-Chitosan Nanoparticles against the Biofilm of *Staphylococcus aureus*

**DOI:** 10.3390/antibiotics11101403

**Published:** 2022-10-13

**Authors:** Jiaman Xu, Quan Lin, Maokun Sheng, Ting Ding, Bing Li, Yan Gao, Yulong Tan

**Affiliations:** 1Special Food Research Institute, Qingdao Agricultural University, Qingdao 266109, China; 2Qingdao Special Food Research Institute, Qingdao 266109, China; 3Marine Science and Engineering College, Qingdao Agriculture University, Qingdao 266109, China; 4Marine Science Research Institute of Shandong Province (National Oceanographic Center of Qingdao), Qingdao 266071, China

**Keywords:** chitosan, nanoparticle, antibiofilm, cinnamaldehyde, *Staphylococcus aureus*

## Abstract

Food contamination caused by food-spoilage bacteria and pathogenic bacteria seriously affects public health. *Staphylococcus aureus* is a typical foodborne pathogen which easily forms biofilm. Once biofilm is formed, it is difficult to remove. The use of nanotechnology for antibiofilm purposes is becoming more widespread because of its ability to increase the bioavailability and biosorption of many drugs. In this work, chitosan nanoparticles (CSNPs) were prepared by the ion–gel method with polyanionic sodium triphosphate (TPP). Cinnamaldehyde (CA) was loaded onto the CSNPs. The particle size, potential, morphology, encapsulation efficiency and in vitro release behavior of cinnamaldehyde–chitosan nanoparticles (CSNP-CAs) were studied, and the activity of CA against *S. aureus* biofilms was evaluated. The biofilm structure on the silicone surface was investigated by scanning electron microscopy (SEM). Confocal laser scanning microscopy (CLSM) was used to detect live/dead organisms within biofilms. The results showed that CSNP-CAs were dispersed in a circle with an average diameter of 298.1 nm and a zeta potential of +38.73 mV. The encapsulation efficiency of cinnamaldehyde (CA) reached 39.7%. In vitro release studies have shown that CA can be continuously released from the CSNPs. Compared with free drugs, CSNP-CAs have a higher efficacy in removing *S. aureus* biofilm, and the eradication rate of biofilm can reach 61%. The antibiofilm effects of CSNP-CAs are determined by their antibacterial properties. The minimum inhibitory concentration (MIC) of CA is 1.25 mg/mL; at this concentration the bacterial cell wall ruptures and the permeability of the cell membrane increases, which leads to leakage of the contents. At the same time, we verified that the MIC of CSNP-CAs is 2.5 mg/mL (drug concentration). The synergy between CA and CSNPs demonstrates the combinatorial application of a composite as an efficient novel therapeutic agent against antibiofilm. We can apply it in food preservation and other contexts, providing new ideas for food preservation.

## 1. Introduction

Foodborne diseases are prominent health problems. Due to the long food chain and complex pathogen sources, food is easily contaminated. Meat, poultry, eggs, aquatic products, dairy products and other foods are very vulnerable to *Staphylococcus aureus* [1]. *S. aureus* often contaminates food in the following ways: via food-processing personnel, when food-processing staff or cooks’ hands or clothes are contaminated with germs and are not cleaned; when the food itself is contaminated before processing or is contaminated during processing, resulting in enterotoxin and food poisoning; when the packaging of cooked food products is not sealed and the contents are contaminated during transportation; and when, e.g., dairy cows suffer from suppurative mastitis or there is local purulence in livestock, contamination can occur in other parts of the body, etc. Biofilm can also form on the surfaces of food and food-processing equipment [2]. Bacterial biofilm refers to a large number of aggregated, membrane-like substances formed by bacteria adhering to a contact surface and secreting polysaccharide matrix, fibrin, lipid protein, etc. [3,4]. The extracellular polymers on the surface of the biofilm are intricately aggregated, forming a complex and orderly overall structure, which effectively protects the stability of the biofilm on the carrier surface. The microorganisms in the biofilm occur together in the form of clusters and are firmly adsorbed on the surface of the carrier. Using common physicochemical methods is often difficult to completely remove biofilm. The physical methods include ultrasonic [5,6], low-current and other methods, such as applications of electromagnetic and shock waves. Although physical methods can meet the requirements for removing biofilms, it is difficult to achieve the desired effect for some large-scale processing equipment. The chemical methods include disinfectants and antibiotics, etc. [7]. These common chemical disinfectants, such as sodium hypochlorite and chlorine dioxide, and antibiotics, such as penicillin and tetracycline, cannot achieve ideal removal effects, and their use will also lead to the development of bacterial drug resistance [8]. Therefore, new drugs to overcome biofilm resistance and eliminate biofilm-protected bacteria need to be developed.

Cinnamaldehyde (CA) is an active component extracted from cinnamon bark, which has a variety of effects, such as anticancer, antifungal and antibacterial activities [9,10,11]. It has been reported that CA can repress bacteria, yeasts and filamentous molds by inhibiting ATPases, cell-wall biosynthesis and by changing membrane structure and integrity [11]. Albano et al. confirmed that CA had the ability to reduce the growth of *Staphylococcus epidermidis* in planktonic state, inhibit biofilm formation and eradicate formed biofilm [12]. Yu et al. found that *Campylobacter* strains treated with 15.63 µg/mL CA exhibited significantly decreased bacterial auto-aggregation, motility, exopolysaccharide production and soluble protein levels, which proved that it had the ability to remove biofilm [13]. CA also had an inhibitory effect on *Candida albicans* by inhibiting the release of its virulence factors [14]. Kot et al. found that trans-cinnamaldehyde is a promising antibiofilm agent for use in MRSA-biofilm-related infections [15].

Since the antibacterial properties of essential oils (EOs) are limited by their high volatility and water-insolubility, the stability of EOs has become a key issue in recent years [16]. Nanoparticles (NPs) have good biocompatibility and can be more stable for the release of EOs and have been widely used in encapsulating EOs [17]. NPs loaded with an EO can protect the EO from the external environment to prolong its inhibitory effect on microorganisms. Chitosan (CS) is an alkaline polysaccharide containing more free amino groups. It is used as a biological carrier, having good safety and biocompatibility, and has been identified as having an effective antibacterial membrane effect [18,19,20,21,22]. When the pH is lower than 10, the amino groups will be protonated and positively charged. The lipid layer on the surface of the biofilm is negatively charged and easy to electrostatically adsorb with positively charged chitosan particles [23]. As a pharmaceutical carrier, chitosan nanoparticles (CSNPs), with the advantages of slow or controlled drug release, can improve drug solubility and stability, thereby enhancing drug effects and reducing the side effects of drugs [24]. This paper mainly studied the antibiotic film effect of chitosan nanoparticles with cinnamaldehyde. 

## 2. Results

### 2.1. Characterization of Nanoparticles

The surface morphologies and morphologies of CSNPs were shown by TEM. CSNP-CAs had smooth surfaces and nearly spherical shapes (Figure 1). The average diameters of the CSNPs and CSNP-CAs were estimated to be 167.3 nm and 298.1 nm, respectively. The surface zeta potentials of the CSNPs and CSNP-CAs were (+34.60) mV and (+38.73) mV, respectively. The average amount of CA on the carriers was 39.7%. 

### 2.2. In Vitro Release Studies

It can be seen from the figure (Figure 2) that the release effect of CSNP-CAs can be significantly divided into two stages. The drug release was rapid in the early stage, including the release of unencapsulated CA adsorbed on the surface of the nanoparticles. The drug release was slow in the later stage, which was due to the gradual release of the drug inside the NPs. The cumulative release amount was about 77.01% in 7 h, and the release tended to be gentle after 10 h. The cumulative release amount reached 86.00% in 48 h.

### 2.3. Antibacterial Activity of CA and CSNP-CAs

The antibacterial activity of CA against *S. aureus* strains (NCTC 8325, RN6390, 15981, Col) were evaluated in vitro by measuring MIC and MBC values. The MIC and MBC values for this strain were 1.25 mg/mL and 2.5 mg/mL when the MIC of CSNP-CAs was 2.5 mg/mL.

### 2.4. Inhibition Activity on Biofilm Formation of CA and CSNP-CAs

As can be seen in Figure 3, biofilm reduction was achieved with all the concentrations of CA tested in this work. Both free CA and CSNP-CAs showed good biofilm inhibition. The activities of free CA and loaded CA both increased with the increase in CA concentration, indicating that the inhibition activities of free CA and loaded CA with respect to biofilm were concentration-dependent. For free CA, there was almost no biofilm formation when the concentration was greater than 1.25 mg/mL. In the same concentration range, the inhibitory activity of loaded CA (CSNP-CAs) was slightly lower than that of free CA.

### 2.5. Antibiofilm Activity on Mature Biofilm of CA and CSNP-CAs

In order to evaluate the effect of CSNP-CAs on the established biofilm, we assessed the antibiofilm effect of CA and CSNP-CAs by repeated treatment for two days. Figure 4 shows the antibiofilm activity of CSNP-CAs and CA on mature biofilm at various CA concentrations after 48 h of treatment. As can be seen from the figure, the biofilm was removed at each concentration, but the effect was different. The removal effect of the CSNP-CAs was better than that of free CA. When the drug concentration reached 4 × MIC, the clearance rate of CSNP-CAs for the biofilm was 48.10% and that of free CA was 38.66%.

### 2.6. Antibiofilm Effect of CSNP-CAs on Food-Grade Silicone

The biofilm grown on food-grade silicone platelets was observed by SEM (Figure 5). *S. aureus* biofilm grown without CA treatment exhibited the typical 3D morphology with dense structures and water channels (Figure 5A). Under the same conditions but treated with free CA and CSNP-CAs, the biofilm showed structural changes (Figure 5B–C). CA can be removed by bactericidal action to form biofilms (Figure 5B). The loading of CA and NPs enhanced the removal of biofilms (Figure 5C). The silicone surface showed more blank areas, with only single or no cells sticking to the surface.

The images obtained with CLSM (Live/Dead staining) confirmed the results. In the control group (Figure 6A), a thick green structure (live cells) could be seen, indicating a mature biofilm structure with active cells. Free CA treatment (Figure 6B) resulted in a reduction in biofilm thickness. Biofilms had less biomass and reduced biofilm thickness, and more biofilms were stained red (dead cells). After the treatment with CSNP-CAs (Figure 6C), biofilms had less biomass and reduced biofilm thickness, the biofilm was almost completely red (dead cells) and the structure of the biofilm was destroyed. In conclusion, the CSNP-CAs could better destroy the structure of biofilm and kill the cells.

### 2.7. Effect of CA on Cell-Wall Integrity of S. aureus

As shown in Figure 7A, the content of extracellular alkaline phosphatase in the bacterial solution treated with CA was increased by more than threefold. The content of alkaline phosphatase reached the maximum at about 8 h, and it tended to be stable after 8 h. It could be inferred that CA could destroy the cell wall of *S. aureus* and increase the permeability of the cell wall. 

### 2.8. Effect of CA on Membrane Permeability of S. aureus

Figure 8 clearly shows the effect of CA on the morphology of *S. aureus.* In the control group without CA, the bacterial cells were spherical and relatively intact, and the bacterial cell membrane was smooth and not broken (Figure 8A). When the concentration of CA was 1 × MIC, *S. aureus* lost its original spherical state, many cells collapsed and cell membranes even ruptured (Figure 8B). After adding CA, the permeability of cell membranes increased, leading to the outflow of potassium ions and proteins in cells. In Figure 7B, when CA was added to the bacterial solution, the concentration of extracellular potassium ions increased gradually and reached a peak at 5 h.

Meanwhile, the concentration of protein in the bacterial suspension also increased gradually (Figure 7C). Compared with the control, the protein content increased six times. This result showed that the addition of CA could rupture the cells of *S. aureus* and cause cytoplasmic leakage. 

The damaged membranes of *S. aureus* cells were observed by flow cytometry, as shown in Figure 9. The figure shows three different types of cells, namely, FITC-/PI- living cells (Q3), FITC+/PI+ dead cells (Q2) and FITC-/PI+ necrotic cells and debris (Q1). It can be seen from the figure (Figure 9A) that almost all control cells were clustered in the Q3 type, which proves that *S. aureus* has a complete cell-membrane structure in this state. However, after CA treatment (Figure 9B), the majority of cells changed from Q3 to Q1, which revealed that the integrity of *S. aureus* membranes was disrupted after CA treatment. This phenomenon is consistent with the results observed in SEM experiments. All experiments proved that cell membranes were damaged after CA treatment.

## 3. Discussion

Nowadays, there is an urgent need to develop a method that can efficiently solve the biofilm problem of foodborne pathogens without causing drug resistance in pathogens. NPs are defined as particles or materials at the nanometer scale [25]. The antimicrobial agent loaded on NPs can be protected from sequestering drugs by the biofilm matrix [26]. The antibacterial agent CA selected in this experiment is a safety additive recognized by the Food and Drug Administration (FDA) [27]. The maximum experimental amount used in this paper (5 mg/mL) was much lower than the maximum addition amount (286 mg/mL) specified by the FDA. CA can inhibit the synthesis of bacterial cell walls and affect membrane permeability and enzyme systems. In the experiment, the drug concentration of 1 × MIC led to a significant increase in the content of alkaline phosphatase, which proved that CA can destroy the integrity of bacterial cell walls. The destruction of the bacterial cell walls led to the death of the bacteria, which was consistent with the conclusion of Shreaz et al. [11]. Therefore, this article supports the method of loading CA on CSNPs to clear biofilm and kill bacteria.

Regarding drug delivery, in vitro release profiles for CA from NPs were evaluated. CSNP-CAs showed a typical release profile, with rapid-burst release at the beginning, followed by sustained release. The initial burst may have been due to the drug loading on the surface of the material, after which a persistent pattern was established due to the release of the drug from the NPs. This phenomenon is more effective for biofilm clearance, as higher initial drug doses reduce the antibiotic resistance of surviving bacteria in biofilms [28]. 

CSNP-CAs were not as effective in inhibiting biofilm formation as free CA, precisely because the CSNPs had a sustained-release effect, the drug concentration released in the time period considered here was small and the antibiofilm effect was weak. As shown in this work, CSNP-CAs were more effective in removing biofilms than free CA. The unique antibiofilm properties of chitosan were mainly attributed to its polycationic nature, conferred by the amino functionality (NH_2_) of the N-acetylglucosamine unit [29,30,31]. The positive charge of chitosan reacts electrostatically with negatively charged biofilm components, such as EPS, proteins and DNA, resulting in an inhibitory effect on bacterial biofilms [32,33]. At the same time, nanoparticles can also carry drugs into biofilm and release drugs to act on the bacteria themselves and kill bacteria. Therefore, CSNP-CAs are more effective in scavenging biofilms than free CA.

This experiment has confirmed that CSNP-CAs have the effect of removing biofilm and that the antibacterial effect of cinnamaldehyde plays an important role in it. CA destroyed the cell wall of *S. aureus* and changed the permeability of the cell membrane, resulting in the outflow of potassium ions, alkaline phosphatase, protein and other substances. Alkaline phosphatase is an enzyme between the cell wall and cell membrane. Under normal circumstances, its activity cannot be detected outside the cell. When the cell wall or cell membrane is damaged, alkaline phosphatase will penetrate the outside of the cell and threaten the integrity of the cell wall. Inoue et al. found that potassium-ion leakage provided direct evidence of membrane damage [34]. The content of extracellular potassium ions in the bacterial solution treated with 1 × MIC of CA increased continuously, which clearly showed the damage to the bacteria caused by CA. Since most of the proteins in the cell are present in the cytoplasm, leakage of the cytoplasm leads to increased levels of extracellular proteins. Propidium iodide (PI) is a nucleic-acid dye that cannot penetrate an intact cell membrane. Since the membranes of injured cells and necrotic cells are destroyed, PI can enter the cell membranes of injured cells and necrotic cells [35], so that the cells can be stained with red fluorescence [36]. Phospholipid serine (PS) exists only on the inner side of the cell membrane in normal cells. When a specific injury occurs, the PS acid on the inner side of the plasma membrane is redistributed, flipped from the inside of the cell membrane to the outside of the cell membrane, and exposed on the outer surface of the cell. PS eversion can be detected by Annexin V. Double staining with FITC-labeled Annexin V and PI enables sensitive detection of live, injured and dead cells via flow cytometry. Therefore, the intracellular PI content can verify the permeability of the cell membrane. The changes in the permeability of the cell membrane and plasma membrane of *S. aureus* lead to cell death.

In summary, we have verified that CA can destroy the cell membrane: on the one hand, it can change the cell-membrane potential and cause the leakage and death of potassium ions, proteins and other contents; on the other hand, CA acts on the cell membrane to redistribute PS inside the plasma membrane, which damages the cell membrane. SEM images can also clearly show changes in cell morphology and even cell rupture. In addition, the combination of CSNPs and CA makes it easier for CA to enter the mature biofilm, to achieve the effect of clearing biofilm.

## 4. Materials and Methods

### 4.1. Bacterial Strains and Growth Media

At present, most *S. aureus* strains have drug resistance. We used four different strains (NCTC 8325, RN6390, 15981, Col) for the determination of MIC. *S. aureus* RN6390 can easily form biofilm and has been wildly used for bovine mastitis infection assays in experiments [37,38], so we used RN6390 to conduct the following detailed experiments. The strain was cultured overnight in Tryptic Soy Broth (TSB) medium from Solarbio (Beijing, China) at 37 °C, 220 rpm. 

CS (low-molecular-weight; degree of deacetylation: 75–85%) and CA (≥95%) were purchased from Macklin (Shanghai, China). Other reagents were purchased from Solarbio (Beijing, China).

### 4.2. Preparation of Nanoparticles

CSNPs were prepared by the ion crosslinking method with polyanionic sodium triphosphate (TPP). In brief, CS was dissolved in acetic acid (1%) and stirred overnight to bring the concentration of chitosan to 1 mg/mL. Then, TPP was dissolved in deionized water for 2 h. TPP solution was mixed dropwise with the CS solution under stirring conditions (volume ratio CS: TPP = 3:1). CSNPs were separated by ultracentrifugation (12,000 rpm, 30 min) and then resuspended in PBS. 

### 4.3. Preparation of CSNP-CAs

For the preparation of the CSNP-CAs, CA (1%, *v*/*v*) was dissolved in ethanol (25%, *v*/*v*) and slowly dropped into the prepared chitosan solution under mechanical agitation to ensure that the volume ratio of CA to CS solution was 1:4. Then, the TPP solution was added to the mixed solution to ensure that the volume ratio of CS solution to TPP solution was 3:1. The mixed solution continued to be stirred at 600 rpm for 4 h. The prepared liquid was centrifuged at 12,000 rpm, and the precipitate was washed with distilled water. The CSNP-CA suspensions were freeze-dried and stored at −20 °C before use.

### 4.4. Particle Size and Zeta Potential of NPs

The size and surface charge of the NPs were measured with a ZetaSizer Nano ZS90 (Malvern Instruments, Malvern, UK). CSNP suspensions were analyzed by dynamic light scattering (DLS) for size and potential determinations. The morphological characteristics were confirmed with transmission electron microscopy (TEM, JEM-1200EX, 80kv, Tokyo, Japan).

### 4.5. Drug Encapsulation

The standard curve of CA was determined by UV spectroscopy (Persee, TU-1810, Beijing, China) at 291 nm. The amounts of CA encapsulated in the NPs were determined by centrifugation. The CSNP-CA solution was centrifuged at 12,000 rpm for 30 min. Then, the absorbance of the supernatant was detected by UV spectroscopy. The CA loading capacity (LC) was calculated according to:LC = (A−B)/C(1)
where A = total amount of CA, B = total amount of non-loaded CA and C = weight of the NPs.

### 4.6. In Vitro Release Test

The in vitro drug-release effect of the CSNP-CAs was passed by dynamic dialysis. The CSNP-CAs were transferred into a dialysis bag (with a retained relative molecular weight of 3500) and 50 mL of PBS buffer was used as the release medium. The dialysis bag was kept at a constant temperature of 37 °C and agitated at 70 rpm. At predetermined time intervals, 5 mL of dialysate was taken and supplemented with the same amount of release medium. The samples were measured at 291 nm with a UV spectrophotometer, 3 times for each time point, and an in vitro drug-release curve for the CSNP-CAs was drawn.

### 4.7. Determinations of Minimum Inhibitory Concentration (MIC) and Minimum Bactericidal Concentration (MBC)

The determination of the MIC was carried out according to the method described by Tan et al [3]. Stock solutions of CA were kept in 25% (*v*/*v*) ethanol to enhance their solubility in suspension. Antimicrobial effects of CA were evaluated using a serial twofold dilution method. CA was initially diluted in 25% (*v*/*v*) ethanol (1:100 *v*/*v*) and then in TSB medium. Serial twofold dilutions for CA were prepared to obtain concentrations ranging from 10 to 0.15 mg/mL. A quantity of 100 μL of bacteria at a final concentration of 1 × 10^5^ CFU/mL in TSB was added into the wells of 96-well microplates. Free CA (10, 5, 2.5, 1.25, 0.62, 0.31, 0.15 or 0 mg/mL) was added to each well. The microplate was incubated at 37 °C for 24 h at 150 rpm. MIC refers to the lowest drug concentration that inhibits the growth of bacteria in culture medium after 18 to 24 hours of culture in vitro. MBC refers to the minimum drug concentration capable of killing cultured bacteria after 18 to 24 hours in vitro. 

### 4.8. Growth of Biofilm

Biofilm formation in 96-well microplates was supported and assayed as described previously [39]. The microplates were incubated at 37 °C for 24 h without shaking. After treatment with different concentrations of CA and CSNP-CAs, the wells were washed 3 times with PBS to remove impurities and plankton. Then, each well was stained with 100 μL of 0.1% (*w*/*v*) crystal violet (CV) solution for 15 min. A quantity of 100 μL of 30% (*v*/*v*) acetic acid was used for extraction. After shaking, the absorbance was measured at 570 nm.

### 4.9. Removal of Biofilm

Biofilm was grown in a 96-well microplate without CSNP-CAs for 48 h, as described above. Then, CSNP-CAs and free CA solution with different concentrations were added to each well to remove the biofilm. After 24 h, the microplates were washed with PBS, and the removal ability was detected by CV staining.

### 4.10. Antibiofilm Effect on Food-Grade Silicone Surfaces

Scanning electron microscopy (SEM) and biofilm-formation analyses were conducted according to the method of Tan et al. [4]. Biofilms were formed and treated with CA and CSNP-Cas, as described above, on silicone (7 mm in diameter; Hongxiang, Jiaxing, China) as the surface substrate. Afterwards, the biofilm was fixed with 3% glutaraldehyde at 4 °C overnight and then dehydrated in a series of ethanol solutions for 20 min each (70%, 80%, 96% and 100%). After chemical dehydration with tert-butanol (Macklin, Shanghai, China), the samples were coated with gold and analyzed via SEM (VEGA3, TESCAN, Brno, Czech Republic), using a 10 kV accelerating voltage. 

### 4.11. Live/Dead Assay

Biofilm cells were stained with a Live/Dead^®^ BacLight™ Bacterial Staining Kit (40274, Yeasen, Shanghai, China), according to the manufacturer’s protocol, for 30 min at 37 °C in the dark. Images of biofilm were analyzed by confocal laser scanning microscopy (CLSM; ZEISS, LSM900, Oberkohen, Germany).

### 4.12. Effect of CA on Membrane Permeability of S. aureus

To the bacterial suspension, 1 × MIC CA was added at 37 °C, and PBS buffer was used as the control. Samples were taken hourly to measure potassium-ion concentration. The determination steps for potassium-ion contents were performed according to the instructions provided with the Nanjing Jiancheng potassium ion kit, and the absorbance was measured at 440 nm (Persee, TU-1810, Beijing, China). All the experiments were carried out in triplicate.

### 4.13. Effect of CA on Cell Wall Integrity of S. aureus

To the bacterial suspension, 1 × MIC CA was added at 37 °C, and PBS buffer was used as the control. Samples were periodically sampled by hourly sampling and centrifuged at 1000 rpm for 10 min, and the supernatant was saved for assay. The instructions provided with the Nanjing Jiancheng alkaline phosphatase determination kit (AKP) were referred to for the determination steps, and absorbance was measured at 520 nm. 

### 4.14. The Morphologies of the Bacteria Observed by SEM

The bacterial suspension was cultured to the logarithmic phase, centrifuged at 4000 rpm for 10 min and the supernatant was discarded. Then, the precipitation was fixed with 2.5% (*v*/*v*) glutaraldehyde at 4 °C overnight, followed by dehydration in a series of ethanol solutions (10%, 30%, 50%, 70%, 80%, 90%, 95%, 100%) for 20 min. Then, the samples were placed in 50% and 100% tert-butanol solutions (Macklin, China) for chemically dehydration for 15 min and vacuum dried for 48 h. Then, the samples were sprayed with gold and analyzed via SEM (VEGA3, TESCAN, Brno, Czech Republic). 

### 4.15. Flow Cytometry Analysis

*S. aureus* was collected by centrifugation (10,000 rpm, 10 min, 4 °C). The bacterial suspension was washed 3 times with sterile PBS buffer and diluted to bacterial suspension OD_600_ = 0.01. Then, 2 × MIC CA was added to the bacterial suspension prior to incubation for 4 h. Flow cytometry analysis was conducted according to the method of Deng et al., with minor modifications [40]. A quantity of 1 mL *S. aureus* bacterial suspension was added to 5 μL of FITC and 5 μL of PI staining solution (Meilunbio, Dalian, China), placed in the dark at 37 °C for 30 min and then detected by flow cytometry (Facs AriaIII, BD, New York, NY, USA).

### 4.16. Statistical Analysis

All the experiments were carried out in triplicate. Statistical analyses were performed with SPSS 19.0 (SPSS Inc., Chicago, IL, USA). Means ± standard deviations (SDs) were calculated for each experiment. Statistical significance was determined by *t*-test analysis, with significant differences determined at *p* < 0.05.

## 5. Conclusions

The results showed that the antibiofilm activity of CA against *S. aureus* was enhanced when loaded on CSNPs. The antibiofilm activity of CSNP-CAs was determined by biofilm inhibition, biofilm disruption and biofilm live/dead bacterial changes on food-grade silicone surfaces. Synthetic CSNP-CAs can be used as potential therapeutics to control *S. aureus* biofilm formation in the future. Furthermore, CSNPs can be used as platforms to design more chemically modified agents or be loaded with other antibiofilm agents for more functional drug-delivery systems.

## Figures and Tables

**Figure 1 antibiotics-11-01403-f001:**
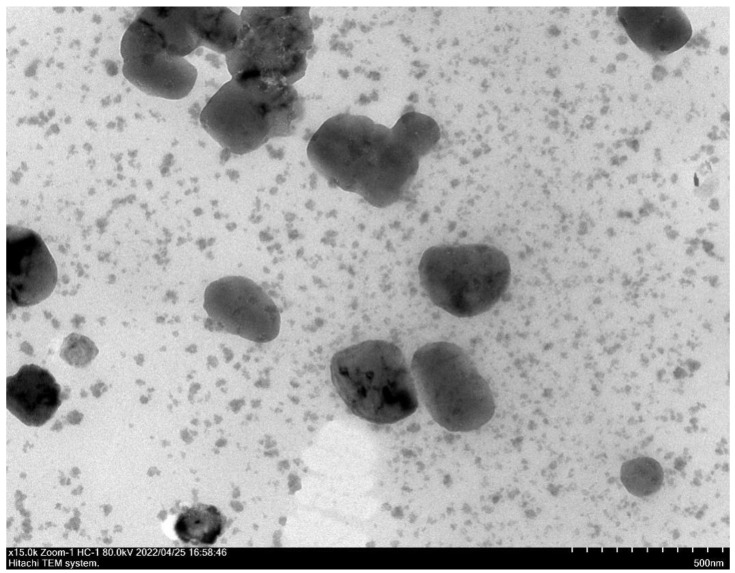
TEM image of CSNP-CAs.

**Figure 2 antibiotics-11-01403-f002:**
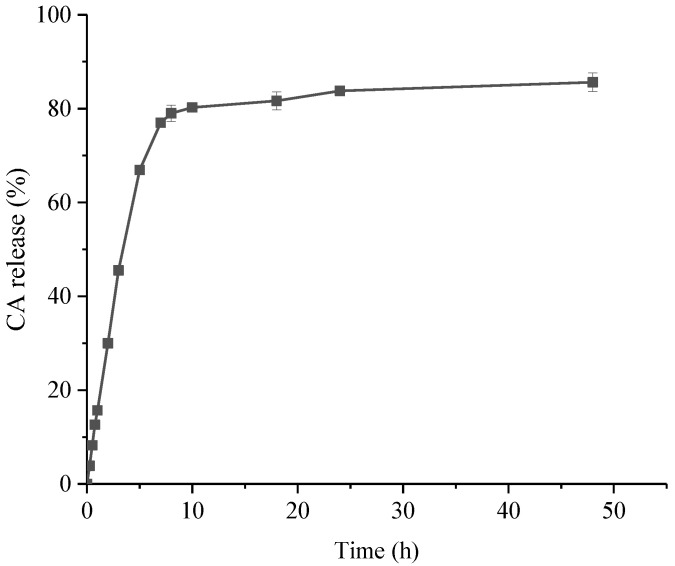
In vitro release profiles for CA from CA-loaded chitosan nanoparticles, using an initial weight ratio of chitosan to CA of 4:1. Data are presented as means ± SDs, *n* = 3.

**Figure 3 antibiotics-11-01403-f003:**
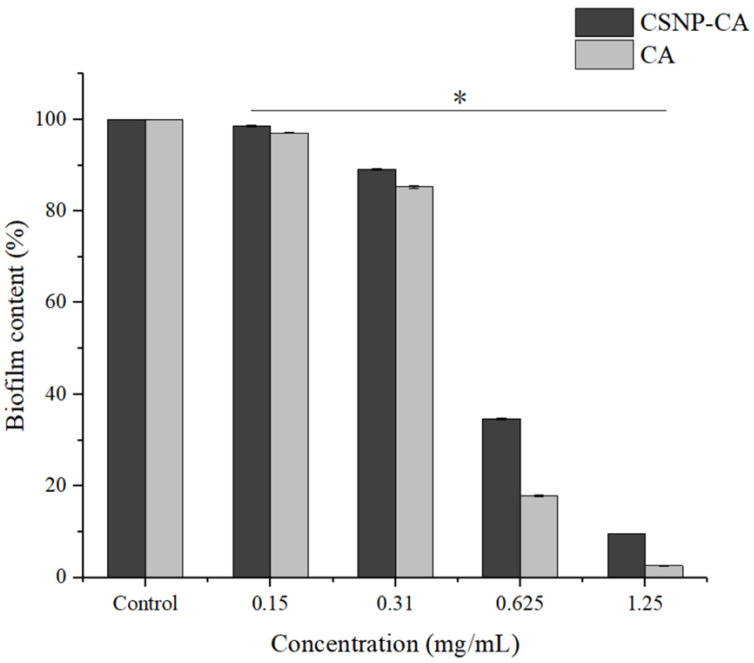
Inhibition effects of NPs on biofilm in 96-well microplates. The results represent the means and standard deviations (error bars) of three independent experiments. * *p* < 0.05 for comparison between the untreated and treated groups.

**Figure 4 antibiotics-11-01403-f004:**
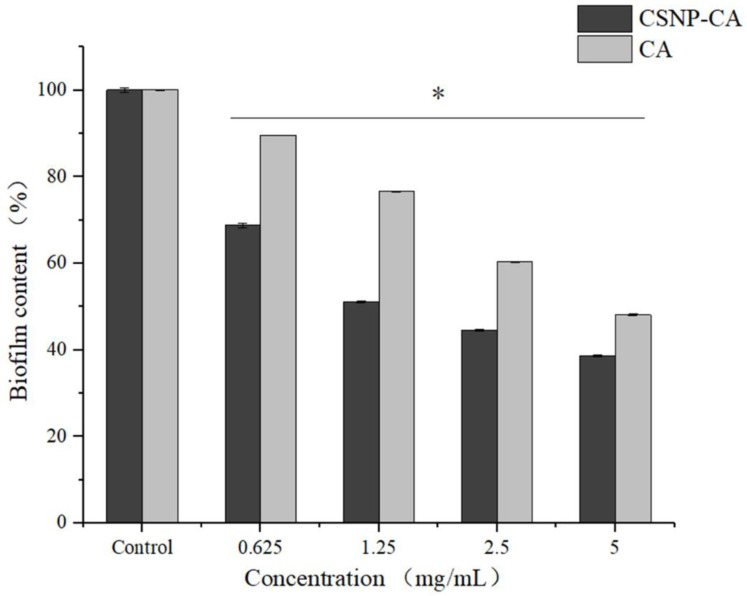
Disruption effects of NPs on mature biofilm in 96-well microplates. The results represent the means and standard deviations (error bars) of three independent experiments. * *p* < 0.05 for comparison between the untreated and treated groups.

**Figure 5 antibiotics-11-01403-f005:**
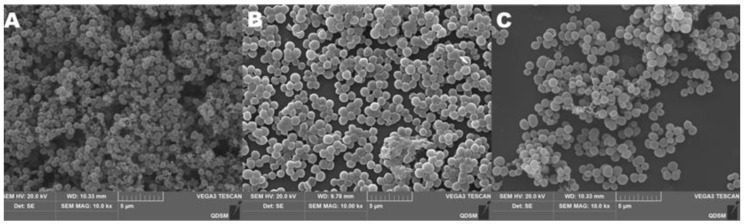
SEM images of *S. aureus* biofilm formations on medical-grade silicone surfaces with media without treatment (**A**) and with CA (**B**) and CSNP-CA treatments (**C**).

**Figure 6 antibiotics-11-01403-f006:**
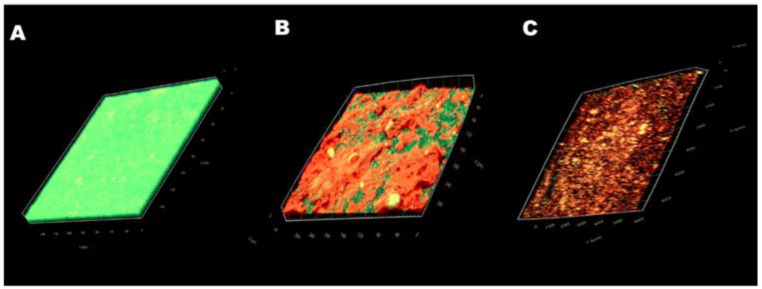
CLSM images of *S. aureus* biofilm formations on medical-grade silicone surfaces without treatment (**A**) and with CA (**B**) and CSNP-CA treatments (**C**). Biofilms were stained with the Live/Dead^®^ BacLight™ Bacterial Viability and Counting Kit. CLSM reconstructions show the three-dimensional staining patterns for live cells (SYTO-9, green) and dead cells (propidium iodide, red). Magnification, ×10.

**Figure 7 antibiotics-11-01403-f007:**
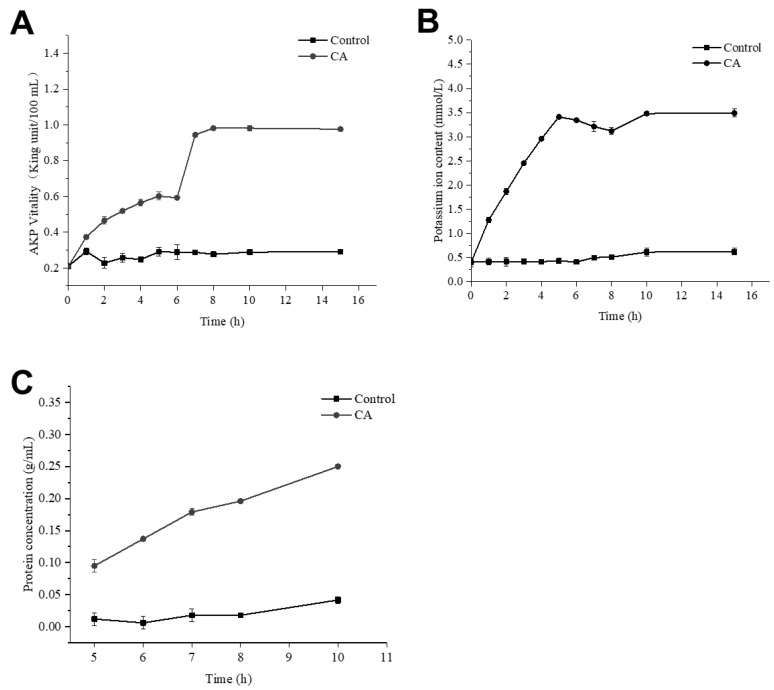
The changes in AKP concentration (**A**), K^+^ concentration (**B**) and protein concentration (**C**) for *S. aureus* treated with CA.

**Figure 8 antibiotics-11-01403-f008:**
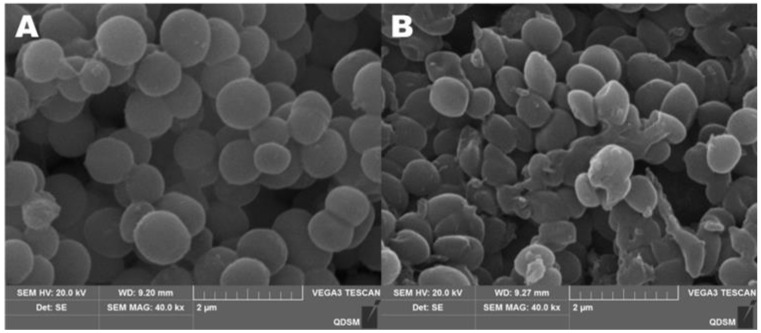
SEM images of *S. aureus* untreated (**A**) and treated (**B**) with cinnamaldehyde for 4 h.

**Figure 9 antibiotics-11-01403-f009:**
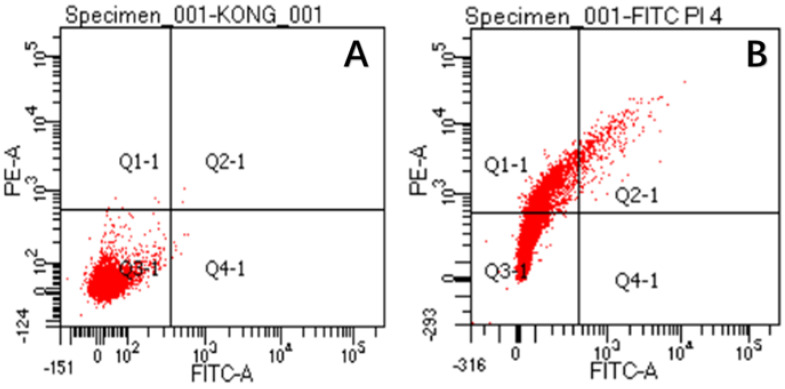
The permeability of *S. aureus* cell membranes before (**A**) and after CA treatment (**B**).

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
