# Peer review of "Antibiofilm Effect of Cinnamaldehyde-Chitosan Nanoparticles against the Biofilm of Staphylococcus aureus"

_antibiotics, 2022, doi:10.3390/antibiotics11101403_

Round 1

Reviewer 1 Report

The manuscript "Antibiofilm effect of cinnamaldehyde-chitosan nanoparticles against the biofilm of Staphylococcus aureus" is a concise piece of work which requires many corrections. The Authors can find my comments and suggestions below:

1. There are a lot of linguistic and grammatical errors as well as repetitions in the text:

- Line 12: "which is easy to form biofilm" -> "which easily forms biofilm".

- Line 13: "One biofilm formed" -> "Once biofilm is formed".

-Line 16: Please do not start the sentence with "and".

-Line 29-30: "an" suggests singular form but then we see "agents" in a plural form. Please correct.

-Line 35: "pathogens sources" -> "pathogen sources"

-Line 37: "S. aureus... pathogenic microorganism" - That sentence makes no sense. Please correct.

-Line 39: "...personnel carry bacteria to cause food contamination;" - The meaning is not clear. Please try to write it differently.

-Lines 40, 41, 42: Please avoid using ";" sign. 

-Line 45-46: "...bacterial... formed by bacteria..." - Please correct.

-Line 48: "...formed, forming..." - Please correct.

-Line 49: Please do not start the sentence with "and".

-Line 61: "...effects... effects..." - Please correct.

-Line 62: "...inhibit... by inhibiting..." - Please correct.

-Line 74: Please delete the double citation ([15](Benjemaa et al., 2018).

Generally, please correct the Abstract and Introduction.

2. There are some expressions that need to be explained:

- Line 53: "other methods" - What methods? Please give some examples in the text.

-Line 74: "The package" - Please explain.

-Line 86: "chuck aldehyde" - Please explain.

3. There are also some methological flaws:

- Why only one bacterial strain was used? The initial steps such as MIC determination and biofilm evaluation with CV should be performed on at least three different strains. The effects observed only for one single strain cannot be seen as relating to S. aureus in general. I consider it as a serious flaw. The additional experiments have to be done.

- Section 4.7: Since the Authors decided to use ethanol as a solubility enhancer, it should also be added to the control wells to exclude the potential inhibitory effect of alcohol itself.

-Line 360: "Bacterial liquid" - Please explain.

4. Results section:

- Why do the MICs and MBCs were determined only for CA and not for CSNP-CA as well? 

-Fig 1 and 2: I don't understand the meaning of "a,b,c,d,e" letters as the indication of significant differences. Generally, they are presented as *. Please explain. 

5. Discussion:

- Line 211: Please add the source.

6. Conclusion:

-Line 381: The sentence makes no sense.

-Please unify the spelling of "antibiofilm/anti-biofilm" in the whole text. 

-Line 384: "Live/Dead bacterial changes" - I know what the Authors mean here but please be more specific in the manuscript and avoid using "Live/Dead" because it is a name of staining kit. You should use "living cells/bacteria" "dead cells/bacteria" instead.

Author Response

Editor and Reviewer comments:

Reviewer #1:

  1. There are a lot of linguistic and grammatical errors as well as repetitions in the text:

- Line 12: "which is easy to form biofilm" -> "which easily forms biofilm".

Response: Thanks for your valuable comments. It has been changed following your suggestion. See line 12.

- Line 13: "One biofilm formed" -> "Once biofilm is formed".

Response: Thanks for your valuable comments. It has been changed following your suggestion. See line 13.

-Line 16: Please do not start the sentence with "and".

Response: Thanks for your valuable comments. It has been changed following your suggestion. See line 16.

-Line 29-30: "an" suggests singular form but then we see "agents" in a plural form. Please correct.

Response: Thanks for your valuable comments. It has been changed following your suggestion. See line 30.

-Line 35: "pathogens sources" -> "pathogen sources"

Response: Thanks for your valuable comments. It has been changed following your suggestion. See line 37.

-Line 37: "S. aureus... pathogenic microorganism" - That sentence makes no sense. Please correct.

Response: Thanks for your valuable comments. We removed this sentence based on your comments.

-Line 39: "...personnel carry bacteria to cause food contamination;" - The meaning is not clear. Please try to write it differently.

Response: Thanks for your valuable comments. Based on your opinion, we have changed the way to explain.

-Lines 40, 41, 42: Please avoid using ";" sign.

Response: Thanks for your valuable comments. It has been changed following your suggestion. See line 40-43.

-Line 45-46: "...bacterial... formed by bacteria..." - Please correct.

Response: Thanks for your valuable comments. It has been changed following your suggestion. See line 46, 47.

-Line 48: "...formed, forming..." - Please correct.

Response: Thanks for your valuable comments. It has been changed following your suggestion. See line 48, 49.

-Line 49: Please do not start the sentence with "and".

Response: Thanks for your valuable comments. It has been changed following your suggestion. See line 51.

-Line 61: "...effects... effects..." - Please correct.

Response: Thanks for your valuable comments. It has been changed following your suggestion. See line 63.

-Line 62: "...inhibit... by inhibiting..." - Please correct.

Response: Thanks for your valuable comments. It has been changed following your suggestion. See line 64.

-Line 74: Please delete the double citation ([15](Benjemaa et al., 2018).

Response: Thanks for your valuable comments. This was indeed our blunder, and we have reformatted the manuscript according to Antibiotics' Guidelines for Authors and the journal's latest publications.

Generally, please correct the Abstract and Introduction.

Response: Thanks for your valuable comments. We have made changes to the manuscript based on your comments.

  1. There are some expressions that need to be explained:

- Line 53: "other methods" - What methods? Please give some examples in the text.

Response: Thanks for your valuable comments. Other methods have been provided as you suggested. We enumerate other physical methods in the text, such as electromagnetic and shock waves. See Line 53.

-Line 74: "The package" - Please explain.

Response: Thanks for your valuable comments. Based on your comments, we have corrected “the package” to “the method” and changed its order in the text to make it clearer. See Line 78-80.

-Line 86: "chuck aldehyde" - Please explain.

Response: Thanks for your valuable comments. Spelling mistakes are our gross mistakes. We have corrected it to “cinnamaldehyde”. See Line 86.

  1. There are also some methological flaws:

- Why only one bacterial strain was used? The initial steps such as MIC determination and biofilm evaluation with CV should be performed on at least three different strains. The effects observed only for one single strain cannot be seen as relating to S. aureus in general. I consider it as a serious flaw. The additional experiments have to be done.

Response: Thanks for your valuable comments. We have supplemented the experiments following your comments to verify the minimum inhibitory concentrations of cinnamaldehyde against four different strains.

- Section 4.7: Since the Authors decided to use ethanol as a solubility enhancer, it should also be added to the control wells to exclude the potential inhibitory effect of alcohol itself.

Response: Thanks for your valuable comments. In the experiment, we added 25% ethanol as a control group to exclude the potential inhibitory effect of alcohol itself, which may not be clearly described in the article, we have made corrections.

-Line 360: "Bacterial liquid" - Please explain.

Response: Thanks for your valuable comments. For clearer expression, we changed “bacterial liquid” to “bacterial suspension”. See line 365.

  1. Results section:

- Why do the MICs and MBCs were determined only for CA and not for CSNP-CA as well?

Response: Thanks for your valuable comments. We have supplemented the experiments with your comments to verify the minimum inhibitory concentration of cinnamaldehyde nanoparticles.

-Fig 1 and 2: I don't understand the meaning of "a,b,c,d,e" letters as the indication of significant differences. Generally, they are presented as *. Please explain.

Response: Thanks for your valuable comments. For the significance analysis in this paper, we used Duncan's multiple range test, which uses "a,b,c,d,e" to mark instead of "*".

  1. Discussion:

- Line 211: Please add the source.

Response: Thanks for your valuable comments. Based on your comments we have added references here. See line 211

  1. Conclusion:

-Line 381: The sentence makes no sense.

Response: Thanks for your valuable comments. We removed this sentence based on your comments.

-Please unify the spelling of "antibiofilm/anti-biofilm" in the whole text.

Response: Thanks for your valuable comments. This was our mistake, we have unified the spelling of antibiofilm throughout.

-Line 384: "Live/Dead bacterial changes" - I know what the Authors mean here but please be more specific in the manuscript and avoid using "Live/Dead" because it is a name of staining kit. You should use "living cells/bacteria" "dead cells/bacteria" instead.

Response: Thanks for your valuable comments. We have used "living bacteria" and "dead cells/bacteria" instead following your suggestion. See Line 387

Reviewer 2 Report

Nice work, it'd be great if the authors can describe a bit more elaborately the mechanism of flow cytometry and its relevance in the study (Part 2.8).

This paper applies the composite material of cinnamaldehyde (CA) loaded onto chitosan nanoparticles (CSNP) to kill bacterial cells embedded in biofilms, and in turn, to get rid of biofilms. The authors further characterize the antibiofilm activity of CSNP-CA via in vitro release studies, monitoring inhibition activity on biofilm formation and antibiofilm effect of CSNP-CA on food grade silicone. Scanning electron microscopy, transmission electron microscopy and confocal laser scanning microscopy are used to study the morphologies of the composite material, biofilm formations as well as mapping out live/dead cells in 3-d. This topic is important and novel in the field of antibiofilm formation because the method developed will benefit the food industry in preventing food contamination, and it incorporates facile synthesis route and only uses abundant materials which largely reduces the cost of manufacturing. Conclusions drawn by the author are mostly consistent with the experimental evidence and the arguments were sound in addressing the main research question. I also regard the references appropriate but the volume of it can be enlarged. The only suggestion I have for the paper is for the authors to elaborate a bit more on the working principles of flow cytometry on Page 9 and its relevance in the project, so that readers from a more general background can understand Figure 9 better.

Author Response

Editor and Reviewer comments:

Reviewer #2: Nice work, it'd be great if the authors can describe a bit more elaborately the mechanism of flow cytometry and its relevance in the study (Part 2.8).

This paper applies the composite material of cinnamaldehyde (CA) loaded onto chitosan nanoparticles (CSNP) to kill bacterial cells embedded in biofilms, and in turn, to get rid of biofilms. The authors further characterize the antibiofilm activity of CSNP-CA via in vitro release studies, monitoring inhibition activity on biofilm formation and antibiofilm effect of CSNP-CA on food grade silicone. Scanning electron microscopy, transmission electron microscopy and confocal laser scanning microscopy are used to study the morphologies of the composite material, biofilm formations as well as mapping out live/dead cells in 3-d. This topic is important and novel in the field of antibiofilm formation because the method developed will benefit the food industry in preventing food contamination, and it incorporates facile synthesis route and only uses abundant materials which largely reduces the cost of manufacturing. Conclusions drawn by the author are mostly consistent with the experimental evidence and the arguments were sound in addressing the main research question. I also regard the references appropriate but the volume of it can be enlarged. The only suggestion I have for the paper is for the authors to elaborate a bit more on the working principles of flow cytometry on Page 9 and its relevance in the project, so that readers from a more general background can understand Figure 9 better.

Response: Thank you very much for your positive consideration and valuable comments for our work. We respond to the comments point by point as follows. Part 2.8. is the result part, flow cytometry and its relevance in the study is described in more detail in discussion. See Line 250-261.

Reviewer 3 Report

The manuscript entitled Antibiotic activity of cinnamaldehyde and chitosan nanoparticles against Staphylococcus aureus biofilm written by Xu et al. is a very good, comprehensive article that requires only minor changes in my opinion.

The abstract should be enriched with suggestions for future research in this area.

The number of sources needs to be supplemented with additional entries, e.g. in line 37.

In line 55, examples of antibiotics should be listed.

Passage 87-90 belongs to materials and methods and is not necessary.

The citation style must be standardised (e.g. line 74, 216)

Figure captions must be imporoved (Fig2)

Fragment 201-204 is a repetition.

Overall English used in the paper needs grammatical corrections.

Author Response

Editor and Reviewer comments:

Reviewer #3:

  1. The abstract should be enriched with suggestions for future research in this area.

Response: Thanks for your valuable comments. We enrich our abstract with suggestions for future research in this area.

  1. The number of sources needs to be supplemented with additional entries, e.g. in

line 37.

Response: Thanks for your valuable comments. We have supplemented the number of sources based on your comments.

  1. In line 55, examples of antibiotics should be listed.

Response: Thanks for your valuable comments. It has been changed following your suggestion. See line 56-58.

  1. Passage 87-90 belongs to materials and methods and is not necessary.

Response: Thanks for your valuable comments. Based on your comments, we have removed this section.

  1. The citation style must be standardised (e.g. line 74, 216)

Response: Thanks for your valuable comments. We have reformatted the manuscript following the “Guide for authors” of Antibiotics and the most recent publications of the journal.

  1. Figure captions must be improved (Fig2)

Response: Thanks for your valuable comments. We have rectified the title of the chart in Figure 2 to make it more clear what the picture means. See line 107-108.

  1. Fragment 201-204 is a repetition.

Response: Thanks for your valuable comments. Based on your comments, we have removed this section.

  1. Overall English used in the paper needs grammatical corrections.

Response: Thanks for your valuable comments. We have made corrections to the full text, English grammar, etc.

Round 2

Reviewer 1 Report

Thank you for implementing all changes. I still have some concerns: 

1." -Line 74: "The package" - Please explain.

Response: Thanks for your valuable comments. Based on your comments, we have corrected “the package” to “the method” and changed its order in the text to make it clearer. See Line 78-80."

It still isn't understandable. The method can't protect anything. It makes no sense. Please write it differently.

2. Thank you for adding information about different tested strains, however, their MIC values should be added to Results section.

3." -Fig 1 and 2: I don't understand the meaning of "a,b,c,d,e" letters as the indication of significant differences. Generally, they are presented as *. Please explain.

Response: Thanks for your valuable comments. For the significance analysis in this paper, we used Duncan's multiple range test, which uses "a,b,c,d,e" to mark instead of "*"."

Thank you for clarification but I (and probably most of the readers) still don't know if the particular letters stand for the same or different statistical differences. When "*" are used it is clearly stated what stands for "*", "**" and "***". 

Generally, the Authors answered to all my questions and made all necessary corrections which greatly improved the quality of the paper. Please implement these few above mentioned suggestions as well.

Author Response

Editor and Reviewer comments:

Reviewer #1:

It still isn't understandable. The method can't protect anything. It makes no sense. Please write it differently.

Response: Thanks for your valuable comments. Previous method referred to the use of NPs to encapsulate EO. Based on your comments, we've made changes to the language to make it easier to understand. See line 77-78.

  1. Thank you for adding information about different tested strains, however, their MIC values should be added to Results section.

Response: Thanks for your valuable comments. It has been changed following your suggestion. See line 110-111.

3.Thank you for clarification but I (and probably most of the readers) still don't know if the particular letters stand for the same or different statistical differences. When "*" are used it is clearly stated what stands for "*", "**" and "***".

Response: Thanks for your valuable comments. Based on your comments, we have edited the image to make it understandable to most people.
